# Silver Nanoparticle Sensor Array-Based Meat Freshness Inspection System

**DOI:** 10.3390/foods12203814

**Published:** 2023-10-18

**Authors:** Jiahang Yu, Mingyuan Huang, Huixin Tian, Xinglian Xu

**Affiliations:** 1State Key Laboratory of Meat Quality Control and Cultured Meat Development, Jiangsu Collaborative Innovation Center of Meat Production and Processing, Quality and Safety Control, College of Food Science and Technology, Nanjing Agricultural University, Nanjing 210095, China; yjh8319@yeah.net (J.Y.); hnzzhmy@foxmail.com (M.H.); yjh@chzu.edu.cn (H.T.); 2School of Biological Science and Food Engineering, Chuzhou University, Chuzhou 239000, China

**Keywords:** colorimetric sensor array, Ag NPs, VOCs, nanosensors, chilled broiler meat

## Abstract

The series of biochemical reactions, metabolic pathways, and regulatory interactions that occur during the storage of meat are the main causes of meat loss and waste. The volatile compounds produced by these reactions, such as hydrogen sulfide, acids, and amines, can directly indicate changes in the freshness of meat during storage and sales. In this study, a one-pot hydrothermal method based on a surface control strategy was used to develop nanoparticles of silver with different reactivities, which were further immobilized in agar powder to develop a colorimetric sensor array. Due to the different chemical interactions with various volatile compounds, the colorimetric sensor array exhibited distinct color changes. The study demonstrates significant differences between 12 different volatile compounds and provides a quantitative and visual method to reveal rich detection indicators. The colorimetric sensor array is an economical and practical multi-analyte identification method. It has many potential applications such as food packaging, anti-counterfeiting, health monitoring, environmental monitoring, and optical filters.

## 1. Introduction

The human olfactory system uses combinations of oral coding to distinguish thousands of volatile compounds (VOCs), although there are only hundreds of receptors [1]. VOCs that enter the nose are detected through olfactory sensory neurons, which transmit signals to the olfactory bulb and onto the brain, thereby allowing the conscious perception of VOCs (Figure 1a) [2]. The application of the human system to meat spoilage detection is problematic owing to its subjectivity and tendency to be fatigued easily, which lead to considerable differences in individual assessments. Consequently, there is a pressing industrial demand for an instrument that can mimic the human olfactory system.

The colorimetric sensor array (CSA) based on a large number of cross-responsive sensor elements is being increasingly used and developed in complex applications such as environmental monitoring [3], food safety detection [4], biomedical diagnosis [5], and explosive detection [6], simulating the olfactory system of mammals. The CSA consists of many sensor elements, which facilitate the detection of more complex odors. When the colorimetric sensor array is combined with individual analytes, it generates various signals and provides unique color patterns for identifying each analyte. With the development of biomimetic technology, rapid detection methods such as the electronic tongue (E-tongue) and electronic nose (E-nose) have been developed for assessing the freshness of meat [7]. However, these methods mainly rely on weak typical van der Waals and physical adsorption interactions, which are easily influenced by environmental conditions such as temperature and humidity. On the other hand, the chemical responsive dyes in the CSA have strong interactions with analytes (e.g., metal–ligand, Lewis acid–base, and π–π interactions), providing excellent detection sensitivity and stability. CSA is more objective, rapid, non-destructive, and not susceptible to interference. CSA-based vapor sensing has become a powerful method in the detection of diverse VOCs [8]. This cross-responsive sensor technology aims to visually quantify multiple VOCs by producing a unique composite response for each VOC, thereby providing a potential assessment of meat safety and quality.

A series of biochemical reactions, metabolic pathways, and regulatory interactions that occur during the storage of meat are the main causes of meat loss and waste. Monitoring spoilage during food storage and transportation is a global problem [9]. During bacterial spoilage, characteristic VOCs such as H_2_S [10], amines (e.g., cadaverine, trimethylamine) [11,12], and acids (e.g., formic acid, lactic acid) [13] are produced. Taking chilled broiler meat as an example, different metabolic products are generated in stored chilled broiler meat due to various biochemical reactions. Enzymes and microorganisms act on fat tissue, producing peroxides, which are then hydrolyzed to produce VOCs such as alcohols, aldehydes, ketones, and low levels of fatty acids [14,15]. Proteins are hydrolyzed to form amino acids, which undergo oxidation–reduction reactions such as dehydrogenation and decarboxylation to form nitrogen-containing organic bases, carboxylic acids, alcohol acids, and other organic decomposition products [16,17]. Among them, amino acids containing cysteine are degraded by enzymes and release sulfur-containing compounds such as hydrogen sulfide [18]. It should be noted that various biochemical reactions can generate typical rancid odors, and it can be assumed that the formation of VOCs is the result of the interaction between different biochemical reactions, making it difficult to assign a specific odor to a single biochemical reaction. Therefore, real-time monitoring and analysis of these characteristic VOCs through a colorimetric sensor array is a potential new method for monitoring food freshness.

By selecting suitable nanoparticles (NPs), a CSA can respond to a range of meat spoilage VOCs. NPs’ ability to self-assemble has been utilized in a wide range of applications. Owing to the composition of their inorganic core, metallic NPs (especially silver and gold) can exhibit strong optical properties [19]. Various molecular ligands (e.g., small molecules [20], carbohydrates [21], cellulose [22], proteins [23], and polymers [24]) have been used to prepare NPs with different physicochemical properties. By varying the surface ligands grafted onto the NP surface, the NPs can have a range of applications.

Silver nanoparticles (Ag NPs) are often used as markers for colorimetric sensors because they have good stability and are easily biofunctionalized. Unlike traditional organic dye-based analysis, Ag NPs have high extinction molar coefficients and size- and composition-dependent localized surface plasmon resonance (LSPR) in the visible regions. Ag NPs with different shapes or sizes can be prepared by changing the synthesis condition [25,26]. Exogenous substances cause LSPR changes due to the aggregation of Ag NPs, accompanied by a distinct color change. The naked eye can easily distinguish these color changes without the help of any instrument. Previous studies introduced ligands on the Ag NP surface with precisely controlled coverage. This stabilized the NPs and enabled effective control of surface charge and particle interactions. Ag NPs were synthesized by using polymers as both surface ligands and reductants, with the addition of small amounts of small molecular ligands as co-reductants. When polymers and small molecule ligands are integrated, combinatorial or synergistic effects may occur and impart new physical or chemical properties to Ag NPs (e.g., colloidal stability [27], mechanical [28], optical emission [29], and catalytic activity [30]).

This study developed an Ag NP-based CSA to monitor a range of VOCs through chemical interactions with analytes. The resulting color changes in a colorimetric gel are easily distinguishable by the human eye. The method represents a significant advance in monitoring meat freshness. The system is cost-effective, non-destructive, selective, sensitive, and accurate.

## 2. Experimental Methods

### 2.1. Materials

Silver nitrate (AgNO_3_), tri-sodium citrate dihydrate (TSC), gelatin (Gel), soy protein isolate (SPI), dextrin (Dex), soluble starch (SS), Casein (Cas), cellulose acetate butyrate (CAB), agar powder (AP), agarose (Aga), guar gum (GG), Konjac (Kon), Xylan (Xyl), sodium dihydrogen phosphate (NaH_2_PO_4_), K_2_HPO_4_·3H_2_O, NaCl, Sodium ferrocyanide (SF), hydrochloric acid (HCl), sodium sulfide nonahydrate (Na_2_S 9H_2_O), sodium hydrosulfide (NaHS), and all of the other flavor substances were purchased from Shanghai Aladdin Bio-Technology Co., Ltd. (Shanghai, China). H_2_S gas was generated by mixing solutions of FeS with dilute sulfuric acid (H_2_SO_4_). Tyvek paper and foam tape (1600T) were purchased from Beijing 3 M Co., Ltd. (Beijing, China). Chilled broiler meat (female, weight, 1000 ± 50 g; age, 14 weeks) was purchased from Jiangsu Lihua Animal Husbandry Co., Ltd. (Changzhou, China). Chilled broiler meat wrapped in modified atmosphere packaging (MAP: 20% CO_2_, 80% N_2_) at 4 °C for 10 days.

### 2.2. Instruments

UV–Vis absorption spectra were measured on a SpectraMax M2 microplate reader (Molecular Devices, Sunnyvale, CA, USA). Chilled broiler meat was packaged with a modified atmosphere packaging machine SMART500 (Ulma Packaging, Guipuzkoa, Spain). Dynamic light scattering (DLS) and ζ-potential measurements were recorded on a Zetasizer nano ZS90 (Zeta sizer Nano, Malvern, UK). Microstructure images were acquired using a TEM microscope (JEM-2100F, JEOL, Tokyo, Japan). Changes in color were recorded by using a mobile phone camera (P30, HUAWEl, Shenzhen, China).

### 2.3. Synthesis of Ag NPs

A polymer was added to ultrapure water (84 mL) and stirred for 10 min. This was followed by the addition of a AgNO_3_ solution (10^−2^ m, 10 mL), a TSC solution (10^−2^ m, 5 mL), and a polymers and small molecular ligands solution (10^−2^ m, 1 mL). The mixture was placed in a high-pressure steam sterilizer for treatment at 130 °C for 25 min. The Ag NP solutions were stored in the refrigerator at 4 °C until use. By changing the TSC and polymers and small molecules, 12 different nanomaterials were prepared (Table 1).

### 2.4. Preparation of Colorimetric Sensor Array (CSA)

#### 2.4.1. Preparation of Colorimetric Gel

Agar powder (1 g) was added to ultrapure water (50 mL) and stirred at 100 °C for 1 h until complete dissolution. This was followed by the addition of a Ag NP solution (50 mL) and stirring at 100 °C for 2 min. Following solidification at room temperature, the gel was cut into 2 mm long cylinders (diameter = 10 mm).

#### 2.4.2. Preparation of Colorimetric Gel Tank

The colorimetric gel tank was made of foam tape with a 10 mm inner ring radius. The bottom was sealed with Tyvek paper, which has excellent breathability (Appendix A).

The colorimetric gel was placed in the colorimetric gel tank to obtain a CSA.

### 2.5. Detection of TVB-N

In order to assess the freshness of chilled broiler meat, the total volatile basic nitrogen (TVB-N) method was employed to measure the quantity of nitrogenous compounds present in the sample. The TVB-N analysis of the chilled broiler meat sample followed the China National Food Safety Standard methods—Method for analysis of hygienic standard of meat and meat products (GB5009.228-2016). Briefly, a 10 g portion of minced chilled broiler meat sample was transferred to a Kjeldahl distillation unit. The volatile biogenic amines were then distilled and collected by utilizing a boric acid solution (20 g/L) containing 5 droplets of a mixed indicator, created by dissolving 0.2 g of methyl red and titrated with a 0.01 M HCl solution. Subsequently, the TVB-N content was determined by calculating the amount of HCl used during titration and expressed as mg/100 g. All experiments were performed in triplicate.

### 2.6. Detection of Gaseous H_2_S

A simple gas phase H_2_S generation device was assembled in the laboratory. The Ag NP solution was added to a 2 mL centrifuge tube (without a cap), and the centrifuge tube was secured in a gas-tight packaging box with double-sided tape. The gas-tight packaging box was then filled with a mixture of 20% CO_2_ and 80% N_2_ and sealed. Afterward, FeS and H_2_SO_4_ solutions were injected into the gas-tight packaging box using a syringe, and the needle hole was sealed. At room temperature, H_2_S gas was generated through the reaction of the FeS and H_2_SO_4_ solution, where the molar concentration of H_2_SO_4_ was three times that of FeS. The excess H_2_SO_4_ in the solution aided in preventing the formed H_2_S from diffusing back into the solution, ensuring the quantitative formation of H_2_S gas.

At 25 °C and 101.325 KPa,
X = M × A/24.45(1)

A: volume concentration unit of gas (ppm);

X: mass concentration unit of gas (mg/m^3^);

M: molar mass of H_2_S;

24.45: volume of grams of gas at 25 °C and 101.325 KPa.

### 2.7. Statistical Analyses

The results are expressed as the mean ± standard deviation. PCA was performed and the Euclidean distance was calculated by using Origin Pro 2021. Tests for significance were conducted using one-way ANOVA in SPSS software (SPSS 16.0, Chicago, IL, USA) (*p* < 0.05).

## 3. Results and Discussion

Twelve types of Ag NPs were prepared that cross-reacted with twelve VOC types and concentrations (hydrogen sulfide, formic acid, acetic acid, 2-methylbutyric acid, lactic acid, cadaverine, putrescine, tyramine, ammonia, methylamine, dimethylamine, and trimethylamine) to produce a distinct color change. The Ag NPs were immobilized in agarose hydrogel to produce a colorimetric gel, with twelve colorimetric gel combinations to cover commercial CSA applications (Figure 1b). The Ag NP surface ligands are food-safe additives with excellent biocompatibility.

The Ag NPs were prepared with twelve different surface ligands, with AgNO_3_, TSC, and polymers with small molecular ligands (prepared through one-pot facile reflux reactions) as the precursor, crosslinker, and surface ligands, respectively. The UV–Vis spectra absorption peaks (400–450 nm) were characteristic of the nanosilver surface plasmon resonance effect and consistent with the absorption peak of a silver nanoparticle solution [31]. (A1-A12, Figure 2a). The Ag NP colors produced with and without the crosslinking action of TSC were dark with obvious characteristic peaks (A1-A7) and light with less distinct peaks (A8-A12), respectively. To better reveal the sensing mechanism, A2 was used as a volatile vapor sensor (H_2_S as a model vapor) for the verification of the reaction in the aqueous phase using NaHS instead of H_2_S [32]. The NaHS aqueous phase reaction was recorded with digital photographs and UV–Vis spectra (Figure 2b). As shown, Ag NPs reacted with increasing NaHS concentrations, changing colors from bright yellow, light brown, dark brown, purple, and finally to light gray. Additionally, the plasmonic peak was red-shifted, indicating NaHS-induced aggregation of the Ag NP particles [33,34]. Simultaneously, the plasmonic peak originally at ∼413 nm gradually decreased in intensity and eventually disappeared. In comparison, the plasmonic peak at ∼288 nm started to develop and eventually evolved into a distinct peak (Figure 2b), indicating that the Ag NPs are gradually etched by NaHS and eventually lose the Ag NP structure. The above conclusions indicate that the aggregation and etching of Ag NPs by NaHS occur simultaneously [35]. The physicochemical properties of the NPs were mainly reflected in their hydrodynamic diameter and ξ-potential. The hydrodynamic diameter of silver nanoparticles was determined using the DLS technique. The results showed that the size of Ag NPs (A2) decreased sharply from 89.95 ± 1.16 nm (0 μM) to 18.33 ± 0.58 nm (100 μM) as the NaHS concentration increased from 0 ppm to 70 ppm (Figure 2c), indicating that the etching effect of NaHS caused the silver atoms on the surface of Ag NPs to be peeled off layer by layer, resulting in a decrease in particle size. As shown in Figure 2d, with the increase in NaHS concentration, the ξ-potential on the surface of Ag NPs changed from −44.56 ± 1.03 mV to −6.52 ± 0.49 mV, indicating that the etching effect of NaHS resulted in a decrease in the surface charge of Ag NPs (A2) and a decrease in ligand coverage density. These results indicate that the sensing principle of this detection method is the aggregation and etching of Ag NPs (A2) caused by the increase in NaHS content, which leads to a significant change in the absorption wavelength intensity and color of the Ag NP (A2) solution, thereby achieving visual detection and UV-visible quantification of NaHS exposure. All of these results indicate that this induction depends on the reaction of NaHS with Ag NPs (A2) to form Ag_2_S, which causes significant color and spectral changes.
2Ag^+^ + HS^−^ = Ag_2_S(2)

However, different exogenous substances can lead to different structural changes in Ag NPs, further verifying the color response of Ag NPs (A2) to acidic substances. As shown in Figure 2e, with the increase in formic acid concentration, the peak intensity of the plasma at ~413 nm in the UV spectrum of Ag NPs (A2) initially decreased and eventually disappeared, but the spectrum did not undergo a red shift or blue shift, indicating that formic acid only caused etching of Ag NPs (A2) without aggregation. Since Ag NPs (A2) do not exhibit a color response to amine substances, it confirms the color response of Ag NPs (A8) to ammonia. As shown in Figure 2f, with the increase in ammonia concentration, the yellow color of Ag NPs (A8) gradually deepened and became visible to the naked eye. At the same time, the plasma peak at ~430 nm became sharper and more symmetric, confirming the further formation of metallic nanoparticles. This indicates that ammonia can induce the further assembly of Ag NPs (A8) to form more stable nanoscale particles. To verify the morphological changes of Ag NPs (A8) during the assembly process, transmission electron microscopy (TEM) was performed on Ag NPs (A8) at different assembly stages (Figure 2g–i). The results showed that with the increase in ammonia concentration, the particle size of Ag NPs (A8) decreased and the distribution became more uniform, indicating that ammonia increased the stability of the Ag NP (A8) system and promoted the assembly reaction of Ag NPs (A8), resulting in a significant color change visible to the naked eye, which can be used for the visual identification of ammonia.

Recently, hydrogels and nanoparticles have been combined to further expand their applications. The construction of CSA is mainly based on chemical-responsive dyes and solid supports. The choice of solid supports has a significant impact on the performance of CSA. Agar powder is used as a solid carrier due to its uniform structure, low cost, and photochemical stability. Thus, the Ag NPs were immobilized in agar powder to produce hybrid hydrogels, which were then placed in a colorimetric gel tank (Appendix A) to fabricate the final CSA.

To verify the sensitivity of the twelve Ag NPs’ color change responses, varying concentrations of VOCs were prepared under liquid phase conditions and mixed with equal amounts of Ag NPs. Like the human olfaction system where one olfactory receptor can respond to different VOCs, each Ag NP responds to a range of VOCs to form a complex gas fingerprint. Each type of VOC can trigger several Ag NPs, and each Ag NP responds to several VOCs, to form a colorful combination of bars that represent the fingerprint for these 12 VOCs. A mobile phone camera (HUAWEI P30) was used to obtain images of the twelve Ag NPs’ color changes after exposure to different concentrations of VOCs (Figure 3). The images were imported into Adobe Photoshop 2020 software, and the average RGB pixel intensities were collected. All RGB images of the CSA were captured four times and averaged.

Although the colorimetric results can be discerned even by the human eye, the data were processed using Euclidean distance (ED) values and linear discriminant analysis (PCA) methods. The similarities between the data clusters of the acquired data (12 Ag NPs × 12 VOCs × 8 concentrations × 4 RGB values) were assessed for VOC concentrations between 10 and 100 µM (Appendix A). The detection limit was 5 µM for NaHS, which is significantly higher than the values for amines and acids (both 10 µM). Using the first three PCs, the PCA 3D score plot shows the separation of the twelve VOC clusters at the same concentration. Additionally, the higher the concentration, the more obvious the separation of the different clusters. Acids and amines data clusters also showed the same trend (Appendix A). ED values of R, G, and B for each Ag NP before and after exposure to all VOCs were calculated (Figure 4). The ED values of the study’s Ag NPs quantitatively describe the concentration of the VOCs. As shown in Figure 4b, the amines’ ED values overlap when the concentration is below 5 ppm but can be discerned separately above 5 ppm. In Figure 4c, for the acids’ ED values, the equivalent concentration is 20 ppm. The study showed that each concentration of VOCs generated specific gas fingerprints. This is because the different chemical groups in the twelve VOCs induce different color combinations across the twelve different Ag NPs.

This single-reaction CSA provides a simple and effective strategy for distinguishing compounds. Compared with previous studies [4,36,37,38,39,40], the CSA has multiple advantages: it (i) possesses a broader detection range; (ii) contains no toxic substances; (iii) can identify a variety of VOCs; and (iv) requires no complex synthetic procedures. Since the surface ligands used are themselves food additives (with excellent biocompatibility and food safety), this novel colorimetric sensing label is nontoxic and eco-friendly. After grafting different ligands, the easily biofunctionalized Ag NPs show distinct color changes for different VOCs. By immobilizing the Ag NPs in an agar powder, the CSA provided a method for multianalyte identification using a single material. A single colorimetric sensor may not provide sufficient colorimetric information to distinguish between VOCs; however, this CSA can achieve both higher chemoselectivity and higher intrinsic sensitivity, thus greatly improving the differentiation of complex mixtures or highly similar compounds.

Although foods undergo complex biochemical changes during storage, it is unnecessary to directly collect hundreds of biochemical substances in practical applications. Moreover, even with complex analytical techniques, it is challenging to accurately detect and differentiate different components in highly complex mixtures. In the food industry, the critical task is to analyze and identify the freshness changes of foods during transportation, storage, and sales. Therefore, highly differentiated fingerprint identification of the entire food mixture is both rapid and cost-effective. Such fingerprint changes are combined with variations in existing quality indicators (such as TVB-N/VOCs) to indicate the freshness changes in the food. To broaden the application of the method proposed in this study, we used the CSA to validate the freshness changes of chilled broiler meat as an example. The CSA was attached to packaging film inside a modified atmosphere packaging box (MAP: 20% CO_2_, 80% N_2_). The colorimetric gel tank protected the colorimetric gel from leaking or contacting the food while remaining in contact with the produced VOCs. The packages were stored at 4 °C, and daily mobile phone (HUAWEI P30) images were taken across a 10-day period without opening the packaging (Figure 5). The distinct color changes were suitable for monitoring the freshness of chilled broiler meat with the naked eye. Seven duplicate experiments showed the same discoloration trends (Appendix A).

In order to classify the color changes in CSA, the color was compared with the freshness characteristic biomarker TVB-N value of chilled broiler meat. In every 100 g of chicken, meat with a TVB-N value ≤15 mg is considered fresh, meat with a TVB-N value of 15–20 mg is edible but must be cooked, and meat with a TVB-N value ≥20 mg is spoiled [41,42]. As shown in Figure 5, each color gel in the CSA exhibited distinct color changes, resulting in different color combinations that clearly indicated the quality changes in chilled broiler meat. Under the same storage conditions, the initial TVB-N value of the meat (per 100 g) was 7.06 mg; it exceeded 15 mg and 20 mg on the 4th and 7th day, respectively (Figure 6). As shown in Figure 5, particularly on the 4th and 7th days, the clear color changes produced indicated the transition of chilled broiler meat from fresh to edible (cooked) to spoiled quality. The CSA developed in this study demonstrated an immediate response to the quality changes in chilled broiler meat. Furthermore, it was observed that each individual colorimetric gel in the CSA could be used separately to indicate the freshness of the meat. Additionally, visual inspection did not show any differences in color for the blank samples (pure water) stored at 4 °C, indicating the CSA’s good stability and selectivity.

The developed CSA responds to changes in the freshness of chilled broiler meat and reliably serves as a freshness history recorder at 4 °C. The CSA generates irreversible color changes and operates on a time scale similar to the existing TVB-N monitoring. The designed CSA has high potential as a real-time freshness monitoring device. Consumers or retailers can visually evaluate the freshness of chilled broiler meat. Additionally, since different foods also produce a range of VOCs during storage, this CSA can theoretically be widely applied to various food packaging systems, such as those used for eggs, dairy products, meats, and vegetables, and may have significant potential implications for improving current food packaging.

## 4. Conclusions

In summary, we have reported a novel and highly diverse CSA based on Ag NPs, which has been developed through various chemical interactions with analytes. In this study, different food-grade ligand molecules were successfully grafted onto the surface of Ag NPs using surface regulation strategies and a one-pot hydrothermal method. The resulting silver nanoparticles with different reactivities were further immobilized in agar powder to obtain the CSA, which was used for the quantitative detection of freshness changes in chilled broiler meat during storage. The CSA can non-destructively detect various VOCs released by stored chilled broiler meat and generate specific color combinations. Optimization algorithms such as ED and PCA were successfully introduced to optimize the color composition of the preprocessed CSA, and these color models could accurately and robustly assess freshness changes in chilled broiler meat, showing a good correlation with the freshness biomarker TVB-N. By modifying the Ag NPs ligands or hydrogel composition, this sensor array can be applied in many fields, including food inspection, safety screening, biomedical diagnosis, environmental monitoring, optical detectors, and wearable electronic devices.

## Figures and Tables

**Figure 1 foods-12-03814-f001:**
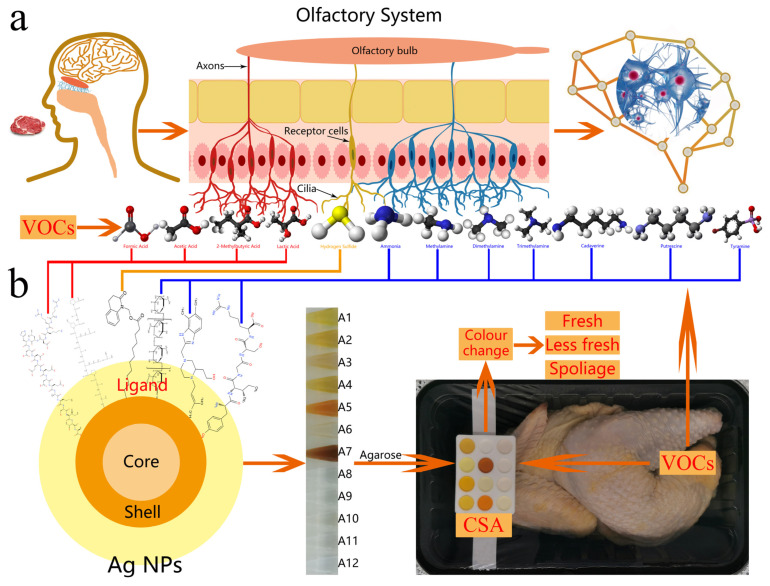
Working principles of (**a**) human olfactory system and (**b**) Ag NP-based CSA system.

**Figure 2 foods-12-03814-f002:**
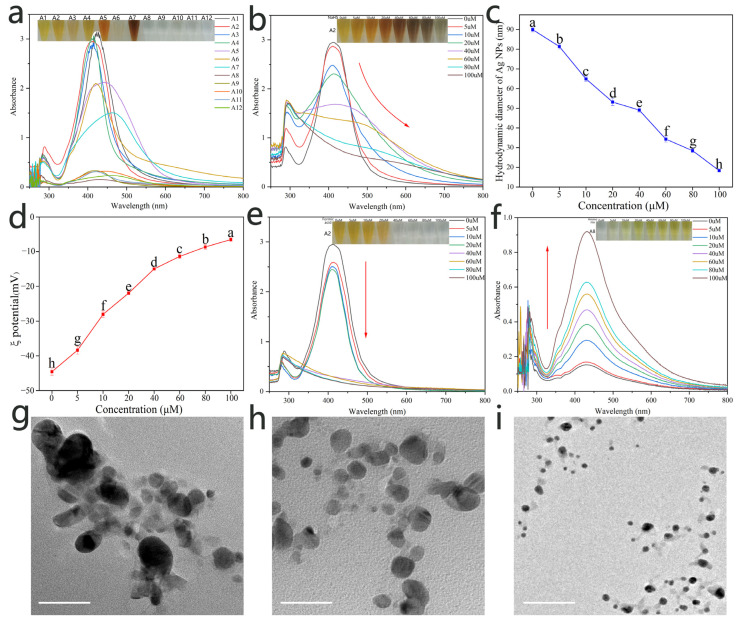
(**a**) Digital photographs and UV–Vis spectra of twelve Ag NPs; (**b**) digital photographs and UV–Vis spectra of Ag NPs (A2) after reaction with increasing NaHS concentrations; hydrodynamic diameter (**c**) and ξ potential (**d**) of Ag NPs (A2) after reaction with increasing NaHS concentrations, different letters mean a significant difference; (**e**) digital photographs and UV–Vis spectra of Ag NPs (A2) after reaction with increasing formic acid concentrations; (**f**) digital photographs and UV–Vis spectra of Ag NPs (A8) after reaction with increasing ammonia concentrations; and TEM image of Ag NPs (A8) after reaction with increasing ammonia concentrations: (**g**) 5 μM, (**h**) 40 μM, and (**i**) 100 μM (scale bar = 50 nm).

**Figure 3 foods-12-03814-f003:**
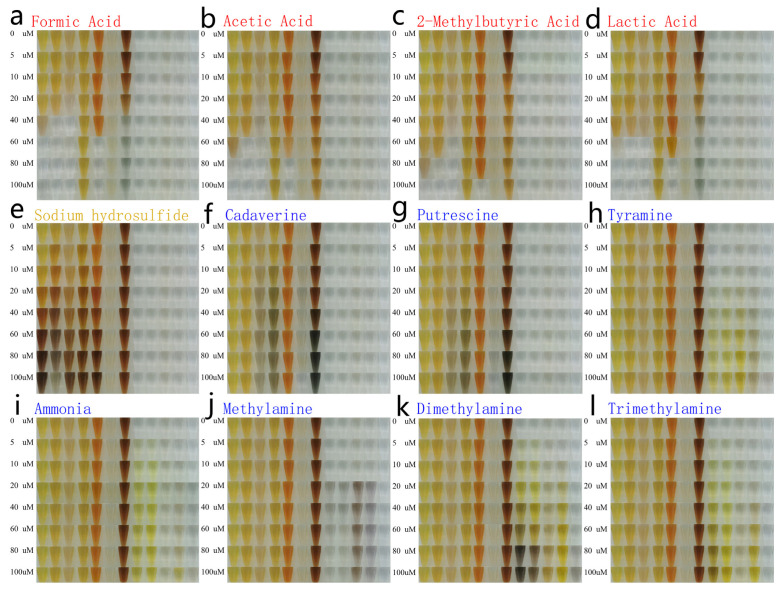
Color changes of twelve Ag NPs after exposure to different concentrations of multiple VOCs.

**Figure 4 foods-12-03814-f004:**
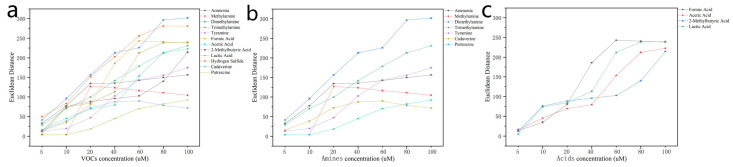
Euclidean distance of R, G, and B values in the twelve different Ag NPs after exposure to different concentrations of (**a**) VOCs, (**b**) amines, and (**c**) acids. The error bars are the standard deviation from four independent experiments.

**Figure 5 foods-12-03814-f005:**
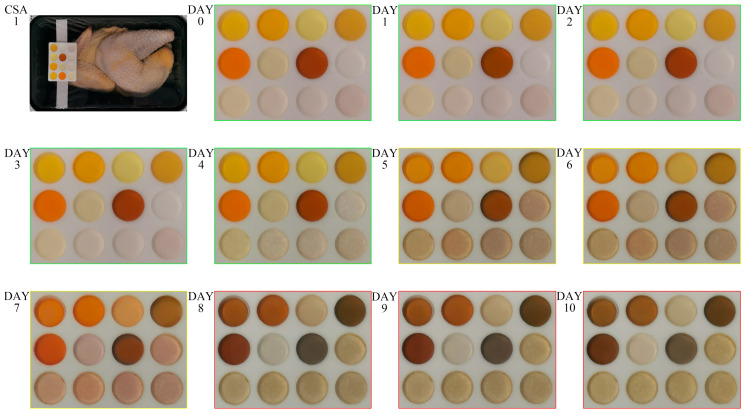
Color change of CSA in chilled broiler meat wrapped in modified atmosphere packaging (MAP: 20% CO_2_, 80% N_2_) at 4 °C.

**Figure 6 foods-12-03814-f006:**
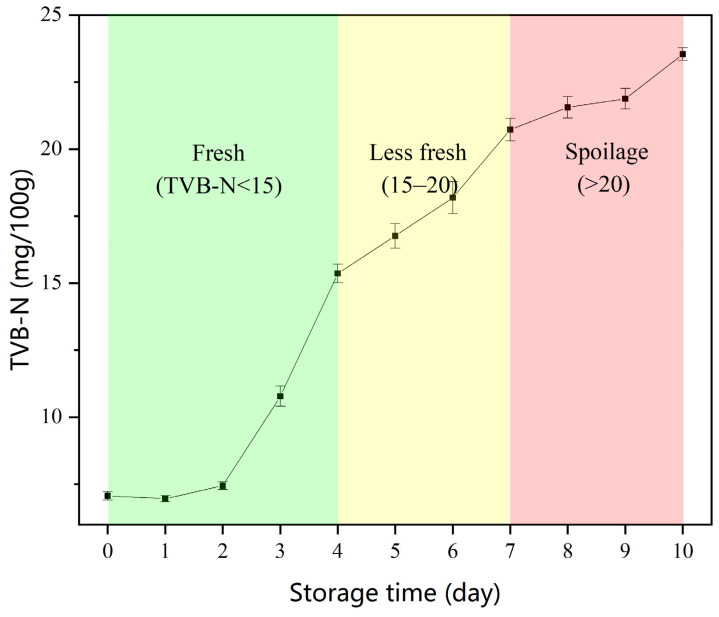
Changes in TVB-N in packaged chilled broiler meat stored at 4 °C.

**Table 1 foods-12-03814-t001:** Composition of different Ag NPs.

Ag NPs	Precursor	Crosslinker	Polymers	Small Molecular Ligands
A1	AgNO_3_	TSC	Gel (0.1 g/100 mL)	NaH_2_PO_4_·2H_2_O
A2	AgNO_3_	TSC	SPI (0.02 g/100 mL)	K_2_HPO_4_·3H_2_O
A3	AgNO_3_	TSC	Dex (0.02 g/100 mL)	NaH_2_PO_4_·2H_2_O
A4	AgNO_3_	TSC	SS (0.5 g/100 mL)	NaH_2_PO_4_·2H_2_O
A5	AgNO_3_	TSC	Cas (0.02 g/100 mL)	K_2_HPO_4_·3H_2_O
A6	AgNO_3_	TSC	CAB (0.02 g/100 mL)	NaCl
A7	AgNO_3_	TSC	CAB (0.02 g/100 mL)	SF
A8	AgNO_3_	-	AP (0.02 g/100 mL)	-
A9	AgNO_3_	-	Aga (0.05 g/100 mL)	-
A10	AgNO_3_	-	GG (0.1 g/100 mL)	-
A11	AgNO_3_	-	Kon (0.1 g/100 mL)	-
A12	AgNO_3_	-	Xyl (0.5 g/100 mL)	-

## Data Availability

Data is contained within the article or Appendix A.

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
