# Peer review of "Silver Nanoparticle Sensor Array-Based Meat Freshness Inspection System"

_foods, 2023, doi:10.3390/foods12203814_

Round 1
Reviewer 1 Report
This paper deals with the development of silver nanoparticle sensor array-based meat freshness inspection system. The paper is interesting and I recommend for it with some minor issues.
1. M and M the authors should stated were did they acquire the chicken used in the experiment. Detection of TVBN must be better explained.
2. M and M the author must better describe the experiment, (time of the storage etc.) .This must be added in M and M.
Author Response
Response to Reviewer 1 Comments
Dear Editor and Reviewer,
Thank you very much for your email on 6 October 2023 with which you sent us the valuable comments concerning our manuscript entitled “Silver Nanoparticle Sensor Array-Based Meat Freshness Inspection System” (ID: foods-2660497). The comments are very helpful for revising and improving our manuscript. We have addressed all the comments as shown in the revised manuscript which we hope will now meet with your approval. Revised portions of the manuscript are marked in red.
Reviewers’ comments:
Point 1: M and M the authors should stated were did they acquire the chicken used in the experiment. Detection of TVBN must be better explained.
Response 1: Thank you for your comments, The relevant parameters of chilled broiler meat and its packaging method have been supplemented in the Materials and Methods section (line 100-102 and line 104-109). We also provide a detailed description of the TVB-N detection method (line 126-134).
Point 2: M and M the author must better describe the experiment, (time of the storage etc.). This must be added in M and M.
Response 2: Thank you for your comments. We have improved the Materials and Methods section, reorganized the method for producing CSA (line 116-121), and added a schematic diagram of the gel tank (Figure S1). The packaging method, storage temperature, and storage time of chilled broiler meat have been included in the Materials and Methods section (line 100-102 and line 104-109).
Special thanks to you for your good comments and suggestions! We have tried our best to improve the manuscript. These changes will not influence the content and framework of the paper. All the changes have been marked in red in the revised paper. We appreciate the efforts of the Editors/Reviewers, and hope that the corrections will meet with approval.
Thank you and best regards!
Yours sincerely,
Xinglian Xu
Reviewer 2 Report
This paper addresses a critical issue of meat wastage due to microbial spoilage and proposes a novel solution through a colorimetric sensor array. By utilizing silver nanoparticles and agar gum, the array undergoes color changes in response to various volatile organic compounds. The capability to differentiate among twelve distinct compounds, despite lacking specificity for each individually, showcases its potential for complex mixtures. Additionally, the array's cost-effectiveness and applicability in various fields like food packaging, anti-counterfeiting, and environmental monitoring highlight its versatility. However, more extensive validation and real-world application testing may be needed to fully assess its practicality and reliability.
Recommendations:
- Expand the abstract to provide more context on the sensor's composition, design, and the scope of VOCs it targets.
- Elaborate on the materials and methods section to ensure transparency in the experimental procedures, allowing for reproducibility.
- Enhance the discussion section by providing deeper insights into the significance of the results, potential applications, and comparison with existing technologies.
- Consider redistributing content to maintain a balanced structure across abstract, materials and methods, results, discussion, and conclusion sections.
Author Response
Response to Reviewer 2 Comments
Dear Editor and Reviewer,
Thank you very much for your email on 6 October 2023 with which you sent us the valuable comments concerning our manuscript entitled “Silver Nanoparticle Sensor Array-Based Meat Freshness Inspection System” (ID: foods-2660497). The comments are very helpful for revising and improving our manuscript. We have addressed all the comments as shown in the revised manuscript which we hope will now meet with your approval. Revised portions of the manuscript are marked in red.
Reviewers’ comments:
Point 1: Expand the abstract to provide more context on the sensor's composition, design, and the scope of VOCs it targets.
Response 1: Thank you for your comments. We have rewritten the abstract and included relevant information on the composition, design, and target VOCs range of the sensor as requested (line 13-23).
Point 2: Elaborate on the materials and methods section to ensure transparency in the experimental procedures, allowing for reproducibility.
Response 2: Thank you for your comments. We have further improved the Materials and Methods section, including the selection and packaging method of the chicken material (line 100-102 and line 104-109), the production method of CSA (line 116-121), and the detection method of TVB-N (line 126-134).
Point 3: Enhance the discussion section by providing deeper insights into the significance of the results, potential applications, and comparison with existing technologies.
Response 3: Thank you for your comments, As requested, we have supplemented the importance of CSA, potential applications (line 277-284), and comparisons with existing technologies (line 241-248) in the Discussion section.
Point 4: Consider redistributing content to maintain a balanced structure across abstract, materials and methods, results, discussion, and conclusion sections.
Response 4: Thank you for your comments. We have reorganized the entire manuscript, rewritten the abstract and conclusion sections, and focused on enhancing the Introduction and Discussion sections.
Special thanks to you for your good comments and suggestions! We have tried our best to improve the manuscript. These changes will not influence the content and framework of the paper. All the changes have been marked in red in the revised paper. We appreciate the efforts of the Editors/Reviewers, and hope that the corrections will meet with approval.
Thank you and best regards!
Yours sincerely,
Xinglian Xu
Round 2
Reviewer 2 Report
No comments